# Multi-Indicators and Evidence of Cytotoxicity—A Case Study of a Stream in Central Brazil

**Raylane Pereira Gomes** [1], **Thais Reis Oliveira** [1], **Aline Rodrigues Gama** [1] , **José Daniel Gonçalves Vieira** [2] **and Lilian Carla Carneiro** [1,*]

[1] Laboratory of Biotechnology of Microorganisms, Institute of Tropical Pathology and Public Health, Federal University of Goiás, Goiânia 74605-020, Goiás, Brazil
[2] Laboratory of Environmental Microbiology and Biotechnology, Institute of Tropical Pathology and Public Health, Federal University of Goiás, Goiânia 74605-020, Goiás, Brazil
* Correspondence: carlacarneirolilian@gmail.com

**Abstract:** (1) Background: Aquatic systems are important to the community and the environment, requiring careful assessment, including the monitoring of their waters. Cities are usually built close to aquatic systems, which serve as a source of water for the entire population. With the uncontrolled increase in cities, aquatic environments receive a great pollutant load. (2) Methods: In this context, the present study aimed to evaluate water contamination, evaluating multi-indicators, cytotoxicity and mutagenicity and conducting a multivariate analysis on the João Leite stream in central Brazil. (3) Results: It was demonstrated, by means of multi-indicators of water quality, that according to the CONAMA classification, current Brazilian legislation and the purpose of the João Leite stream, the water quality met some parameters (i.e., turbidity, dissolved oxygen, and bacteriological); however, in some samples, the quality was poor or very poor. Samples collected in the rainy season indicated cytotoxicity, probably due to pollutants dragged by the rain into the stream. Based on multivariate and association analysis, we suggest that the João Leite stream presents anthropogenic pollution. (4) Conclusions: This study provides data for the development of prevention, control and environmental management policies. In addition, we demonstrate that the use of multivariate statistical analyses can provide data on water pollution, its source of pollution and the association between pollutants.

**Keywords:** water pollution; toxicological parameters; aquatic biomonitoring; Cerrado; physicochemical and microbiological water quality





## 1. Introduction

The quality of water in the area of the source is important for human health [1]. Additionally, people also rely heavily on riparian systems for their socioeconomic development [2]. The classification of water quality and its quality standards are described based on comparisons of the limit values (maximum and minimum) of the concentration of specific pollution parameters; these limits are defined by legal instruments or competent national and international guidelines and are based on in the water's use [3]. This classification indicates the water quality, which serves as a basis for controlling pollution [4].

In aquatic biomonitoring programs, multi-indicators (chemical, biological and physical parameters) are evaluated in the water body, but based on analyses of chemical effects, are also used to supplement water quality data [5,6].

The physicochemical characteristics of water influence the functioning of the water body, both at biotic and abiotic levels, such as its primary productivity, trophic structure, food chain, and the determination of ecosystem structures and their species [7]. Therefore, physicochemical parameters are used to obtain the levels of pollution and degradation of

water bodies, determining their quality; this helps in the diagnosis and future conservation/preservation of this environment, as it is a useful methodology to determine the levels of substances present, such as metals, pesticides and others [7–9].

The physicochemical nature of a water body can be influenced by spatial, environmental and climatic factors, weathering, erosion, pollution, runoff from urban and agricultural areas, sewage, and industrial and domestic effluents [10].

The toxicological and ecotoxicological risks of the aquatic environment and their evaluation is extremely important to verify the effects on representative organisms, considering the reach of these risks from cellular structures to individuals, populations and communities, and simulating the possible effects that occur in the aquatic biota [11]. The toxic effects observed on surface waters are not measurable by only quantifying the compounds described in the regulations [12]. Several bioassays using biomarker responses can be performed to assess the ecotoxicological effects of environmental sources [13].

Aquatic ecotoxicology is a science that aims to help with the problems of the contamination of water bodies, signaling their ecotoxicological potential and their mechanisms of action in living organisms. Controlling the toxicity of residues released into the aquatic environment is extremely important for the health of ecosystems and humans, and ecotoxicological tests are scientific proof of environmental changes [14]. Therefore, ecotoxicological tests provide data on the toxic effects of contaminants on aquatic biota, and thus, enable the use of applications and measures to preserve the environment and the organisms that share it [15].

According to [16], in Brazil, there is a lack of studies regarding the ecotoxicity of Brazilian aquatic systems, mainly regarding the protection of native Brazilian species, due to the fact that the country has vast biodiversity. Biomarkers are recommended by ecotoxicologists as an effects-based tool, making it possible to verify the effects of substances (pure and mixtures) and, as a result of their specificity, also discriminate their toxic roles in the biological system [17].

The João Leite stream is an important tributary of the Meia Ponte River, which belongs to the hydrographic basin of the Paranaíba River. Its course runs through the municipalities of Ouro Verde, Campo Limpo, Anápolis, Goianápolis, Terezópolis de Goiás, Neropólis and Goiânia. Its area is equivalent to approximately 764 km$^2$, with its length being equal to 86 km; its climate type is AW (Köppen classification), it has a predominance of oxisols and it is part of the Cerrado biome. This surface water system is responsible for urban supply, and it fulfils demands for irrigation, fish farming and others [18,19]. Despite having three conservation units in its area—the Altamiro de Moura Pacheco Ecological Park, Ipês Park and the João Leite Environmental Preservation Area [20]—approximately 70.25% of the area of this aquatic environment is anthropized [21]. The João Leite basin is affected by an intense degradation process; this is due to agricultural and industrial wastes and also to the activities of urban expansion in the metropolitan region of Goiânia, such as deforestation, the production of punctual pollutants, landfills, landfills effluents, and soil drainage and runoff [22].

In this context, the objectives of this study were: (i) to evaluate the multi-indicators, including physical, chemical and bacteriological parameters and the main metals in the water of the João Leite stream; and (ii) to evaluate the cytogenotoxic effect of the mixture of pollutants in João Leite stream water samples, determining the genetic damage that the supposed pollutants can cause.

## 2. Materials and Methods

### 2.1. Sample Location

The water samples were collected and stored until further analysis according to the guide for the collection and conservation of water samples, sediments, aquatic communities and liquid effluents of the Environmental Company of São Paulo State [23]. One collection was carried out in the dry period and one in the rainy period (November and April, respectively) in 2017. The samples came from the four sampling points shown in Figure 1.

In total, eight samples were collected from each period—one for each sampling point. At each point, samples were collected (approximately 5 liters of water) and divided into several flasks according to [23] to perform all the tests and determine the parameters.

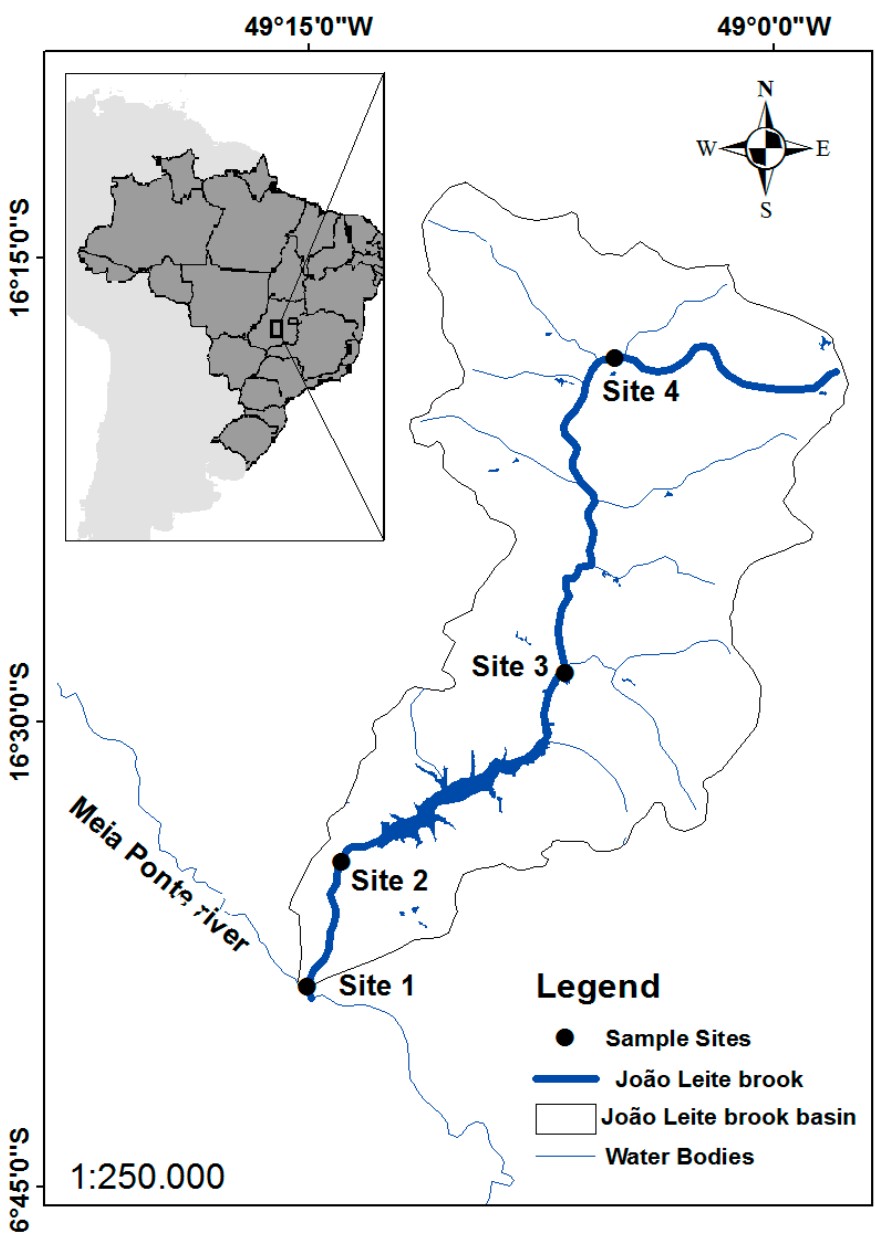

**Figure 1.** Map of the location of the sampling sites of the João Leite stream, Goiás, Brazil. Geographic coordinates—site 1: 16°38′32.91″ S, 49°15′1.97″ W; site 2: 16°34′30.54″ S, 49°13′55.02″ W; site 3: 16°28′25.05″ S,49°6′43.87″ W; site 4: 16°18′13.88″ S,49°5′6.16″ W.

### 2.2. Analysis of Water by Multi-Indicators

Five liters of water were collected at each site to analyze the multi-indicators that correspond to physicochemical parameters and bacteriological examination. The water samples were sent to Companhia Saneamento de Goiás (SANEAGO) and were analyzed using the methods established by the Standard Methods for Examining Water and Wastewater [24]. The samples were processed within 24 h of collection. The parameter values were analyzed and classified according to the Brazilian environmental regulation Resolution 357/2005 of the National Environment Council (CONAMA) [25], which classifies freshwater bodies into

four classes (I, II, III and IV) and a special class—aimed at preserving the natural balance of communities and aquatic environments—that was not used for classification in this study.

### 2.3. A. Cepa Test

To verify the cytotoxicity and genotoxicity of the water samples from the sampling sites, the *A. cepa* test was performed using 5 bulbs for each sample, carrying out the methodology according to [26,27], with modifications suggested by [28].

The external scales and the central parenchyma of the sprouting crown of the organic *A. cepa* bulbs were removed for each sample. The bulbs were cleaned in tap water for 20 min, and then, in distilled water for 60 min. The root area was placed on a display with the water samples in covered glass vials to prevent the passage of light. The samples were exposed for seven days at room temperature, protected from direct sunlight, and the samples' absorbed volumes were replaced every day, twice a day, with the respective samples that were stored at 4 °C. After seven days of exposure, to the roots were measured in millimeters (mm) from the end of the apical meristem of the root to the primordial root plate. With the data on root growth, the relative growth index (*RGI*) (Equation (1)) and the inhibition index (Ii) (Equation (2)) were calculated, according to [29].

$$RGI = \frac{average\,root\,length\,in\,samples}{average\,root\,length\,in\,the\,negative\,control} \tag{1}$$

$$Ii = 100\% - ICR\% \tag{2}$$

To make the slides, the root tips (1–2 cm) were used to count and observe the cells. The root tips were fixed in Carnoy's solution (3: 1 acetic acid p.a./ethanol p.a.) for 6–12 h and washed in distilled water for 5 min. Subsequently, the root tips were heated for 60 s in an acetic orcein solution (2% orcein and 45% acetic acid p.a.) and chorded by approximately 4–5 mm. The blades were prepared viacrushing between the blades and cover slips.

The cell counts and analysis were conducted using an objective $100\times$ optical microscope, with 5000 cells counted per sample. The cells were counted and analyzed by dividing them among cells in mitosis (in division), in inter phase, and with chromosomal and nuclear changes. After counting the cells, the following calculations were performed: mitotic indexes (*MI*) (Equation (3)) to assess the levels of cytotoxicity, and the indexes of chromosomal and nuclear alterations (*CAI*) (Equation (4)) for the evaluation of genotoxic and mutagenic levels, according to [28,30].

$$MI = \frac{number\,of\,cells\,in\,mitosis}{total\,number\,of\,cells\,observed} \times 100\% \tag{3}$$

$$CAI = \frac{number\,of\,cells\,changed}{total\,number\,of\,cells\,observed} \times 100\% \tag{4}$$

The ecotoxicological analysis was also compared with CONAMA Resolution N° 357 [25]. It established that for classes I and II, there can be no chronic toxic effect on organisms; for class III, no acute toxic effect can be obtained; and for class IV, this parameter is not delimited.

### 2.4. Statistical Analysis

For the statistical tests, the Stat Soft software STATISTICA® version 10.0 was used for descriptive statistics (mean and standard deviation). The Student's T test for normal distributions, the Wilcoxon test for other distributions, and correlation analysis and principal component analysis (PCA) were used. For the construction of the heat map, the values obtained using the STATISTICA software was inserted into the Microsoft Excel software (Microsoft Corp., Redmond, WA, USA). A significance level of $p < 0.05$ was adopted for all analyses.

## 3. Results and Discussion

### 3.1. Water Quality by Multi-Indicators

Tables 1–3 show the physicochemical and bacteriological characterization, respectively, found in the samples of raw water from the João Leite stream, Goiás, in addition to indicating the maximum values of references allowed by Resolution CONAMA N° 357 [25].

The statistical test was applied to verify if there was a statistical difference between the dry and rainy periods. Among all the physicochemical parameters, only the parameters of chlorides, nitrite and biochemical oxygen demand were statistically different between periods ($p < 0.05$). This difference can be attributed to the seasonality of the collections. Like in [31], we can attribute the climatic conditions to the differences between the sampling periods in the water analyses of the studied rivers. The rainy season has a greater correlation with urban areas, whereas the dry season has a greater correlation with all types of land use, including agricultural and urban areas [32]. In study [33] conducted in the Taizi River basin, in China, it was indicated that in the rainy season, pollution from point and non-point sources predominates, and in the period of drought, pollution from point sources predominates. This is probably due to precipitation and runoff in the rainy season [34] that carry pollutants into the water of the water bodies. During the drought period, there is a greater concentration of pollutants due to the reduction in water volume, leading to a greater effect of pollutant mixture in the rainy season and greater transport of substances by rainwater [35]. These studies suggest differences in the peroxide found in drought and rain, explaining what may be happening with the parameters that had variations between periods.

In study [36], it was indicated that the João Leite stream can be classified according to CONAMA class II. Observing the isolated results for each sample site (Table 1), we find parameters classified in CONAMA as classes III and IV. The turbidity parameter can be classified as class IV in samples from site 3 and 4 in the rainy period. The dissolved oxygen parameter of site 3 in the dry period can be classified as class III, and the samples of sites 2, 3 and 4 in the rainy period do not fit into any of the values of the four classifications. The bacteriological parameter referring to the total coliform index can be classified as class IV, except for the site 2 samples from the dry period, which can be classified as class III. The E. coli index of site 1 samples from the dry period and site 1, 3 and 4 samples from the rainy period can be classified as class IV.

The turbidity parameter showed an increase in all sampling sites in the rainy season (Table 2). In a river, the turbidity is significantly changed during a period of heavy rain due to the increase in suspended sediments [37]. The increase in dissolved oxygen may be linked to a higher concentration of photosynthesis in algae and plants [38]. This may be an explanation for the high rates of dissolved oxygen in the João Leite stream.

The bacteriological analysis showed that the water quality of the João Leite stream is poor when taking into account the total coliforms and *E. coli* indexes (Table 3); their presence was detected in the two periods of the study, indicating the occurrence of fecal contamination. The microbiological analysis of water is based on the concept of fecal indicator bacteria, which are present in human and animal feces, with *E. coli* and enterococci standing out. The sources of fecal bacteria pollution from environmental waters are direct deposits of feces (human and animal), sewage, effluents, wastewater, leaching and runoff from tanks, landfills that store manure, fertilizers (animal manure) used on agricultural land and the impermeable coverage of urban areas [39].

**Table 1.** Chemical characterization of raw surface water samples from the João Leite stream, Goiás State, Brazil.

| Parameters | Dry Period | | | | | Rainy Period | | | | | Maximum Value Allowed by CONAMA N° 357 | | | |
|---|---|---|---|---|---|---|---|---|---|---|---|---|---|---|
| | Site 1 | Site 2 | Site 3 | Site 4 | Mean ± SD | Site 1 | Site 2 | Site 3 | Site 4 | Mean ± SD | Class I | Class II | Class III | Class IV |
| pH | 7.29 | 7.59 | 7.64 | 7.77 | 7.57 ± 0.20 | 7.20 | 7.78 | 7.59 | 7.25 | 7.46 ± 0.28 | 6 a 9 | 6 a 9 | 6 a 9 | 6 a 9 |
| Total alkalinity (mg/L $CaCO_3$) | 54.00 | 55.00 | 58.00 | 65.00 | 58.00 ± 4.97 | 47.00 | 55.00 | 38.00 | 60.00 | 50.00 ± 9.63 | NR | NR | NR | NR |
| Alkalinity $HCO_3$ (mg/L $CaCO_3$) | 54.00 | 55.00 | 58.00 | 65.00 | 58.00 ± 4.97 | 47.00 | 55.00 | 38.00 | 60.00 | 50.00 ± 9.63 | NR | NR | NR | NR |
| Alkalinity $CO_3$ (mg/L $CaCO_3$) | 0.00 | 0.00 | 0.00 | 0.00 | 0.00 ± 0.00 | 0.00 | 0.00 | 0.00 | 0.00 | 0.00 ± 0.00 | NR | NR | NR | NR |
| Alkalinity OH (mg/L $CaCO_3$) | 0.00 | 0.00 | 0.00 | 0.00 | 0.00 ± 0.00 | 0.00 | 0.00 | 0.00 | 0.00 | 0.00 ± 0.00 | NR | NR | NR | NR |
| Total hardness (mg/L $CaCO_3$) | 48.00 | 48.00 | 52.00 | 68.00 | 54.00 ± 9.52 | 40.00 | 48.00 | 32.00 | 52.00 | 43.00 ± 8.87 | NR | NR | NR | NR |
| Organic matter—oxygen consumed (mg/L $O_2$) | 2.10 | 0.80 | 1.10 | 3.20 | 1.80 ± 1.09 | 2.10 | 0.90 | 5.20 | 3.00 | 2.80 ± 1.82 | NR | NR | NR | NR |
| Chlorides (mg/L Cl) | 6.00 | 6.50 | 7.50 | 10.00 | 7.50 ± 1.78 | 2.50 | 6.00 | 1.00 | 4.50 | 3.50 ± 2.2 | 250.00 | 250.00 | 250.00 | NR |
| Carbon gas (mg/L $CO_2$) | 5.62 | 2.87 | 2.70 | 2.24 | 3.36 ± 1.53 | 6.20 | 1.85 | 1.98 | 6.85 | 4.22 ± 2.68 | NR | NR | NR | NR |
| Total dissolved solids (mg/L) | 69.46 | 69.63 | 76.94 | 97.40 | 78.36 ± 13.17 | 57.97 | 64.57 | 43.40 | 74.14 | 60.02 ± 12.92 | 500.00 | 500.00 | 500.00 | NR |
| Nitrate (mg/L $N-NO_3$) | 0.10 | 0.20 | 0.20 | 0.40 | 0.23 ± 0.13 | ND | ND | 0.10 | 0.70 | 0.20 ± 0.34 | 10.00 | 10.00 | 10.00 | NR |
| Nitrite (mg/L $N-NO_2$) | 0.05 | 0.01 | 0.04 | 0.04 | 0.03 ± 0.01 | 0.01 | 0.00 | 0.01 | 0.02 | 0.01 ± 0.01 | 1.00 | 1.00 | 1.00 | NR |
| Total ammoniacal nitrogen pH ≤ 7.5 (mg/L $N-NH_3$) | 3.50 | ND | ND | ND | 0.88 ± 1.75 | 0.54 | ND | ND | 0.22 | 0.19 ± 0.26 | 3.70 | 3.70 | 13.30 | NR |
| Total ammoniacal nitrogen 7.5 < pH ≤ 8.0 (mg/L $N-NH_3$) | ND | 0.13 | 0.30 | 0.48 | 0.23 ± 0.21 | ND | 0.14 | 0.30 | ND | 0.11 ± 0.14 | 2.00 | 2.00 | 5.60 | NR |
| Sulfate (mg/L SO4) | 6.00 | <1 | 2.00 | 14.00 | 7.33 ± 6.11 | <1 | 1.00 | 8.00 | 6.00 | 5.00 ± 3.61 | 250.00 | 250.00 | 250.00 | NR |
| Dissolved oxygen (mg/L $O_2$) | 6.00 | 5.90 | 2.50 | 6.00 | 5.10 ± 1.73 | 5.80 | 7.30 | 8.00 | 6.50 | 6.90 ± 0.96 | 6.00 | 5.00 | 4.00 | 2.00 |
| Biochemical oxygen demand, 5 days at 20 °C (mg/L $O_2$) | 0.50 | 0.40 | 1.20 | 1.00 | 0.78 ± 0.39 | 1.90 | 1.80 | 3.30 | 1.50 | 2.13 ± 0.80 | 3.00 | 5.00 | 10.00 | NR |

Note: SD: standard deviation; ND: not detected; NR: not regulated. Resolution N° 357 from [25].

**Table 2.** Physical characterization of raw surface water samples from the João Leite stream, Goiás State, Brazil.

| Parameters | | Ambient Temperature (°C) | Water Temperature (°C) | Turbidity (uT) | True Color (uH) | Conductivity (μS/cm) |
|---|---|---|---|---|---|---|
| Dry period | Site 1 | 25.5 | 22.9 | 40 | 28.8 | 126.3 |
| | Site 2 | 25.5 | 24.33 | 1.9 | 4.7 | 126.6 |
| | Site 3 | 25.5 | 22.35 | 15 | 22.8 | 139.9 |
| | Site 4 | 25.5 | 25.29 | 75 | 24.6 | 177.1 |
| | Mean ± SD | 25.50 ± 0.00 | 23.72 ± 1.34 | 32.98 ± 32.17 | 20.23 ± 10.65 | 142.48 ± 23.94 |
| Rainy period | Site 1 | 23.1 | 24.9 | 55 | 32.4 | 105.4 |
| | Site 2 | 27 | 26.5 | 8 | 5.1 | 117.4 |
| | Site 3 | 24.5 | 24 | 230 | 52.8 | 78.9 |
| | Site 4 | 23.2 | 24.6 | 240 | 24.4 | 134.8 |
| | Mean ± SD | 24.45 ± 1.82 | 25.00 ± 1.07 | 133.25 ± 119.12 | 28.68 ± 19.75 | 109.13 ± 23.49 |
| Maximum value allowed by CONAMA N° 357 | Class I | NR | NR | 40 | NR | NR |
| | Class II | NR | NR | 100 | 75 | NR |
| | Class III | NR | NR | 100 | 75 | NR |
| | Class IV | NR | NR | NR | NR | NR |

Note: SD: standard deviation; ND: not detected; NR: not regulated. Resolution N° 357 from [25].

**Table 3.** Bacteriological characterization of raw surface water samples from the João Leite stream, Goiás State, Brazil.

| Parameters | | Total Coliform Index (N.M.P. 100 mL) | *Escherichia coli* Index (N.M.P. 100 mL) |
|---|---|---|---|
| Dry period | Site 1 | >24,200.00 | >24,200.00 |
| | Site 2 | 12030 | 60 |
| | Site 3 | >24,200.00 | 1520 |
| | Site 4 | >24,200.00 | 1460 |
| | Mean ± SD | 21,157.50 ± 6085.00 | 6810.00 ± 11,612.94 |
| Rainy period | Site 1 | >24,200.00 | >24,200.00 |
| | Site 2 | >24,200.00 | 28 |
| | Site 3 | >24,200.00 | >24,200.00 |
| | Site 4 | >24,200.00 | >24,200.00 |
| | Mean ± SD | 24,200.00 ± 0.00 | 18,157.00 ± 12086.00 |
| Maximum value allowed by CONAMA N° 357 | Class I | 1000 | 200 |
| | Class II | 5000 | 1000 |
| | Class III | 20,000 | 4000 |
| | Class IV | NR | NR |

Note: SD: standard deviation; ND: not detected; NR: not regulated. Resolution N° 357 from [25].

The Cascavel River, Brazil, also showed unsatisfactory microbiological quality, with a high rate of contamination by total coliform and *E. coli* [40]. In the Coruja/Bonito watershed, the presence of *E. coli* in its waters was above the limit allowed by CONAMA, and the authors indicated that the presence of this pathogen could cause endemic outbreaks in the population consuming this water [41]. These studies corroborate the indexes of coliforms found in the João Leite stream, indicating that bacteriological contamination can cause disease outbreaks of these bacteria in the consuming population.

*3.2. A. Cepa Test*

The results of the cytotoxicity and genotoxicity assessment of the water samples from the João Leite stream using the *A. cepa* test are shown in Table 4. Regarding root growth and its indexes, site 4 was the sample site with the greatest differentiation in relation to

the negative control. Statistically, the root growth values in the dry period for the water samples should not be statistically different from the of the negative control for any of the analyzed sites in the João Leite stream. In the rainy period, they were statistically different in relation to the negative control for the root growth of *A. cepa* for water analysis at site 1 ($p = 0.0051$), site 2 ($p = 0.0009$), site 3 ($p = 0.0332$) and site 4 ($p = 0.0015$). Only in the rainy period (rainy season) did the *A. cepa* test demonstrate cytotoxicity. Therefore, it is not in accordance with CONAMA legislation N° 357 [25], which establishes that there can be no chronic toxic effect.

**Table 4.** Analysis of root growth inhibition relative growth index, inhibition index, mitotic index and index of chromosomal and nuclear changes using the test organism *A. cepa* for samples from the João Leite stream, Goiás State, Brazil.

| Indicator | Dry Period—Water | | | | | Rainy Period—Water | | | | |
|---|---|---|---|---|---|---|---|---|---|---|
| | Site 1 | Site 2 | Site 3 | Site 4 | CN | Site 1 | Site 2 | Site 3 | Site 4 | CN |
| Mean ± SD of RG | 24.00 ± 12.02 | 29.82 ± 14.54 | 22.44 ± 13.06 | 37.84 ± 18.32 | 27.82 ± 12.37 | 43.45 ± 14.37 * | 14.33 ± 2.41 * | 15.57 ± 8.80 * | 45.00 ± 14.93 * | 28.44 ± 15.00 |
| RGI | 0.86 | 1.07 | 0.81 | 1.36 | 0.00 | 1.53 | 0.50 | 0.55 | 1.58 | 0.00 |
| Ii (%) | 13.74 | −7.16 | 19.36 | −36.00 | 100.00 | −52.81 | 49.60 | 45.27 | −58.24 | 100.00 |
| MI (%) | 7.34 | 1.88 | 1.82 | 0.77 | 11.12 | 0.94 | 2.08 | 2.54 | 2.32 | 1.80 |
| CAI (%) | 5.30 | 2.26 | 2.38 | 2.65 | 2.84 | 0.42 | 1.92 | 1.36 | 1.62 | 1.00 |

Note: *: Statistically different from control ($p < 0.05$); SD: standard deviation; CN: negative control; RG: root growth in mm; RGI: relative growth index; Ii: inhibition index; MI: mitotic index; CAI: index of chromosomal and nuclear changes. A negative control was carried out for each period.

All samples had an increase or decrease in root growth when compared to the negative control, as shown by RGI and Ii (Table 4). Study [42], which evaluated the samples from the Sinos river, Rio Grande do Sul State, Brazil, found that all the samples inhibited the root growth of *A. cepa*, which indicates toxicity.

Evaluating the MI in the dry period, the four samples of water analyzed had a decrease in relation to the negative control (MI = 11.12%), with the smallest decrease referring to site 4 (MI = 0.77%). In rainy period, in relation to water analysis, only site 1 (MI = 0.94%) had a decrease in MI in relation to the negative control (MI = 1.80%); the other samples (site 2, site 3 and site 4) had an increase, with site 4 showing the greatest increase (Table 4).

Cytotoxicity can be determined in environmental biomonitoring compared to the negative control by increasing or decreasing the MI, indicating the presence of toxic and cytotoxic compounds. An increase in the MI indicates an increase in cell division, which can be harmful due to uncontrolled proliferation and tumor formation. A decrease in the MI may indicate that the growth and development of the test organism has been affected [43].

A reduction in the MI below 22% in relation to the negative control can cause lethal effects in the study organism [44]. A reduction below 50% has sub-lethal effects [16], indicated by the cytotoxic limit value [45]. Thus, all the sampling points of the drought period and site 1 of the rainy season had lethal effects.

For CAI in the dry period, site 1 (CAI = 5.30%) had an increase in relation to the negative control (CAI = 2.84%), the other sites had a slight decrease. In the rainy period for CAI, the same trend was observed for the MI in the rainy period, where for water, only site 1 (CAI = 0.42) decreased in relation to the negative control (CAI = 1.00%) and the others increased. The mutagenic effect can be observed through the significant increase in the frequency of chromosomal aberrations and micronuclei [46]. No significant mutagenic effect was found in the samples in the João Leite stream, as in the study by [47], which also evaluated the toxicity of the João Leite stream using a multi biomarker in fish. The author also reports that the comet assay was an effective biomarker to identify DNA damage in caged fish, which corroborates our studies.

Regarding nuclear abnormalities and chromosomal aberrations, we found nuclear fragmentation, nuclear damage, chromatin fragmentation, spindle disorders, chromosomal

breaks, frequencies and chromosomal bridges, which are shown in Figure 2. These nuclear abnormalities and chromosomal aberrations indicate that in brook waters, João Leite had substances that exhibited clastogenic and aneugenic action [48].

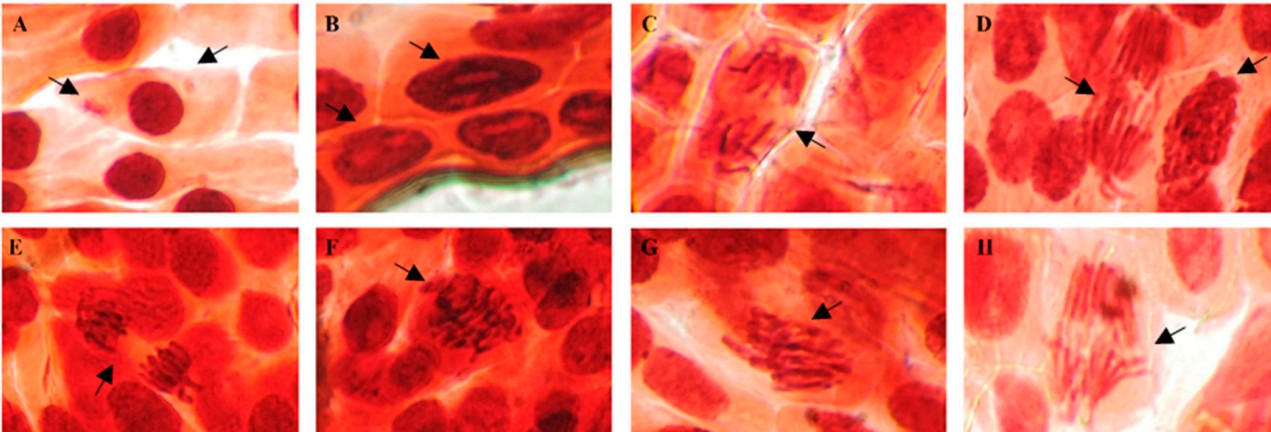

**Figure 2.** Chromosomal and nuclear changes observed in *A. cepa* after exposure to raw and elutriated surface water from sediments from the João Leite stream, Goiás State, Brazil. Photomicrograph taken at 100×. Types of chromosomal and nuclear aberrations observed (indicated by the arrows)in *A. cepa* meristematic cells after 7-day exposure in water and elutriation from the João Leite stream sediment. (**A**) Nuclear fragmentation; (**B**) double nuclear lesion and binucleated cells; (**C**) spindle disorder in anaphase with a chromosomal break; (**D**) chromosomal and lagging bridges, fragments of chromatin; (**E**) sticky anaphase; (**F**) chromosomal errors in metaphase; (**G**) dispersed metaphase sprayed; (**H**) chromosomal fragmentation.

Other samples from other rivers also indicated chromosomal and nuclear anomalies in the countries of Kazakhstan [49], India [50], Thailand [51], Nigeria [52] and Brazil [53]. This indicates that these anomalies are related to pollutants present in the waters.

*3.3. Grouping, Correlation and Analysis of Main Components of the Parameters*

To assess the relationships among the parameters and the samples under study, a statistical analysis was carried out in which the parameters that had no variation or whose detection was zero (alkalinity $CO_3$ and OH, total coliform index and If) were removed. Additionally, potential risk indexes for human health were not included.

Cluster analysis using joining (tree clustering) was performed to verify the relationship between the sampling points and their periods (dry and rainy). From this analysis, it was found that point 1 and point 2 have groupings with their respective collection pairs (dry and rainy periods). On the other hand, location 3 and location 4 are grouped with the periods (Figure 3). Points 3 and 4 are transition areas between the urban and rural environments, location 1 is a total urban area and location 2 is a preservation and rural area; this could be an explanation for the sample groupings of the points. This grouping demonstrates that points 1 and 2 did not undergo major statistical changes in the multiple indicators analyzed in this study, given the climatic changes in the dry and rainy periods. The opposite occurred with points 3 and 4. The multiple indicators used to assess the water quality of the João Leite stream also suggest that in the cluster analysis, the types of pollutants were grouped in the same way and that these locations may have same anthropic influences, since they have similar characteristics. Cluster analysis is highly used by other studies to group sampling sites for aquatic environments based on their similarities [54,55].

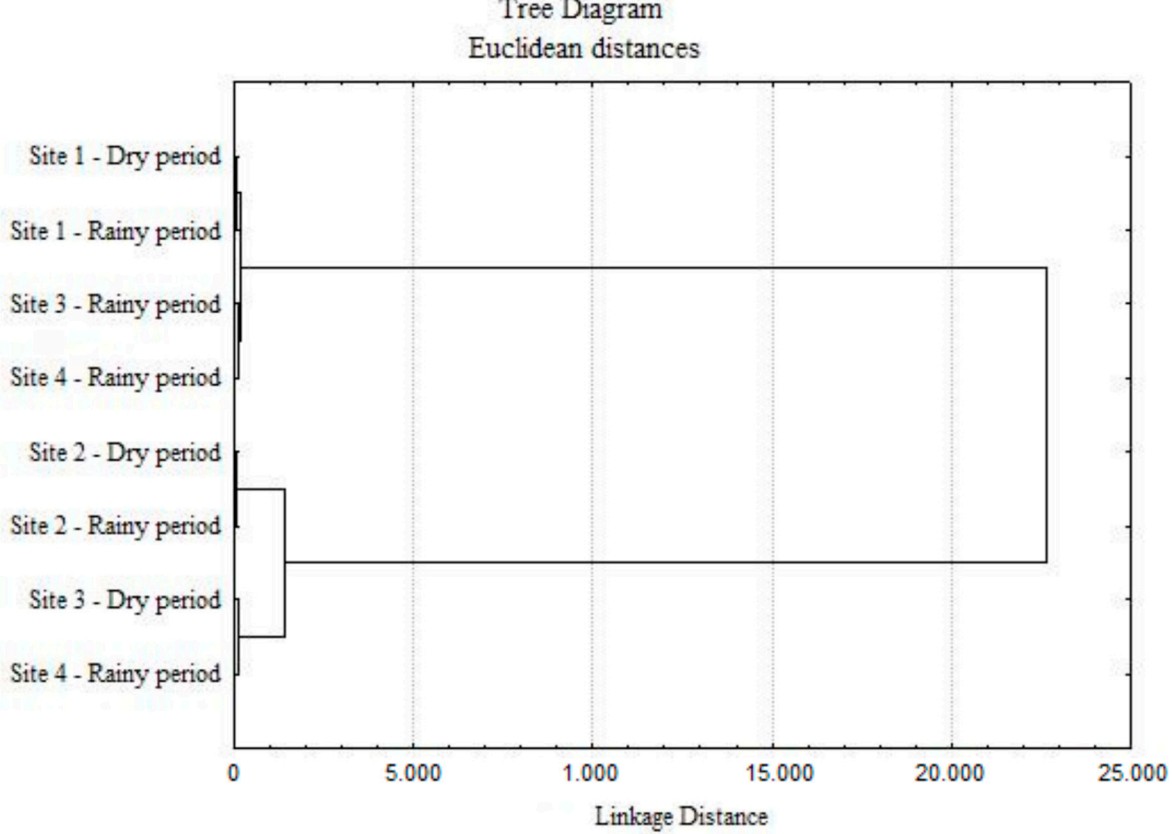

**Figure 3.** Phenogram (Joining Analysis) to demonstrate the relationships between the sample sites studied in the João Leite stream, Goiás State, Brazil.

Pearson's correlation was applied to the 26 multiple indicators of the João Leite stream water analyze. The results obtained by the correlation are shown in Figure 4 in a heat map. The significant correlations ($p < 0.05$) can be called effective parameters [56]. The heat map shows good and strong correlations.

| Variable | TA | TW | Tu | Tc | pH | Alk | Ik HCO3 | TH | OM | CLO | CG | TDS | N-NO3 | N-NO2 | N-NH3 | N-NH38 | CN | SO4 | DO | DBO | Ecl | RG | RGI | Ii | MI | CAI |
|---|---|---|---|---|---|---|---|---|---|---|---|---|---|---|---|---|---|---|---|---|---|---|---|---|---|---|
| TA | 1.00 | 0.14 | -0.62 | -0.54 | 0.78 | 0.27 | 0.27 | 0.30 | -0.45 | 0.57 | -0.73 | 0.24 | -0.38 | 0.22 | 0.04 | 0.38 | 0.24 | -0.05 | -0.05 | -0.30 | -0.74 | -0.70 | -0.70 | 0.70 | 0.17 | 0.47 |
| TW | 0.14 | 1.00 | 0.00 | -0.35 | 0.29 | 0.10 | 0.10 | 0.14 | -0.03 | 0.01 | -0.18 | 0.03 | -0.02 | -0.59 | -0.43 | 0.02 | 0.03 | 0.04 | 0.61 | 0.22 | -0.23 | 0.10 | 0.10 | -0.10 | -0.48 | -0.45 |
| Tu | -0.62 | 0.00 | 1.00 | 0.66 | -0.31 | -0.31 | -0.31 | -0.28 | 0.84 | -0.55 | 0.27 | -0.30 | 0.54 | -0.21 | -0.16 | -0.01 | -0.30 | 0.45 | 0.48 | 0.62 | 0.65 | 0.20 | 0.20 | -0.20 | -0.03 | -0.31 |
| Tc | -0.54 | -0.35 | 0.66 | 1.00 | -0.32 | -0.62 | -0.62 | -0.47 | 0.87 | -0.60 | 0.15 | -0.45 | -0.04 | 0.06 | 0.15 | 0.15 | -0.45 | 0.44 | 0.22 | 0.66 | 0.73 | 0.00 | 0.00 | 0.00 | 0.14 | -0.13 |
| pH | 0.78 | 0.29 | -0.31 | -0.32 | 1.00 | 0.23 | 0.23 | 0.35 | -0.11 | 0.51 | -0.95 | 0.28 | -0.15 | 0.05 | -0.51 | 0.80 | 0.28 | 0.21 | -0.01 | 0.04 | -0.83 | -0.60 | -0.60 | 0.60 | -0.36 | 0.01 |
| Alk | 0.27 | 0.10 | -0.31 | -0.62 | 0.23 | 1.00 | 1.00 | 0.95 | -0.45 | 0.89 | 0.05 | 0.95 | 0.59 | 0.54 | -0.03 | 0.19 | 0.95 | 0.24 | -0.43 | -0.73 | -0.54 | 0.40 | 0.40 | -0.40 | -0.11 | 0.34 |
| Ik HCO3 | 0.27 | 0.10 | -0.31 | -0.62 | 0.23 | 1.00 | 1.00 | 0.95 | -0.45 | 0.89 | 0.05 | 0.95 | 0.59 | 0.54 | -0.03 | 0.19 | 0.95 | 0.24 | -0.43 | -0.73 | -0.54 | 0.40 | 0.40 | -0.40 | -0.11 | 0.34 |
| TH | 0.30 | 0.14 | -0.28 | -0.47 | 0.35 | 0.95 | 0.95 | 1.00 | -0.27 | 0.93 | -0.09 | 0.99 | 0.54 | 0.61 | -0.06 | 0.42 | 0.99 | 0.47 | -0.36 | -0.64 | -0.55 | 0.40 | 0.40 | -0.40 | -0.17 | 0.34 |
| OM | -0.45 | -0.03 | 0.84 | 0.87 | -0.11 | -0.45 | -0.45 | -0.27 | 1.00 | -0.47 | 0.00 | -0.29 | 0.24 | -0.03 | -0.05 | 0.30 | -0.29 | 0.69 | 0.52 | 0.67 | 0.59 | 0.00 | 0.00 | 0.00 | 0.01 | -0.18 |
| CLO | 0.57 | 0.01 | -0.55 | -0.60 | 0.51 | 0.89 | 0.89 | 0.93 | -0.47 | 1.00 | -0.30 | 0.93 | 0.28 | 0.66 | 0.00 | 0.46 | 0.93 | 0.29 | -0.48 | -0.74 | -0.74 | 0.10 | 0.10 | -0.10 | -0.07 | 0.48 |
| CG | -0.73 | -0.18 | 0.27 | 0.15 | -0.95 | 0.05 | 0.05 | -0.09 | 0.00 | -0.30 | 1.00 | -0.04 | 0.33 | 0.04 | 0.48 | -0.78 | -0.04 | -0.14 | -0.05 | -0.19 | 0.71 | 0.70 | 0.70 | -0.70 | 0.31 | 0.04 |
| TDS | 0.24 | 0.03 | -0.30 | -0.45 | 0.28 | 0.95 | 0.95 | 0.99 | -0.29 | 0.93 | -0.04 | 1.00 | 0.54 | 0.66 | -0.03 | 0.39 | 1.00 | 0.44 | -0.46 | -0.69 | -0.52 | 0.40 | 0.40 | -0.40 | -0.16 | 0.36 |
| N-NO3 | -0.38 | -0.02 | 0.54 | -0.04 | -0.15 | 0.59 | 0.59 | 0.54 | 0.24 | 0.28 | 0.33 | 0.54 | 1.00 | 0.25 | -0.20 | 0.07 | 0.54 | 0.46 | -0.06 | -0.21 | 0.07 | 0.60 | 0.60 | -0.60 | -0.15 | -0.01 |
| N-NO2 | 0.22 | -0.59 | -0.21 | 0.06 | 0.05 | 0.54 | 0.54 | 0.61 | -0.03 | 0.66 | 0.04 | 0.66 | 0.25 | 1.00 | 0.52 | 0.31 | 0.66 | 0.50 | -0.54 | -0.57 | -0.11 | 0.00 | 0.00 | 0.10 | 0.45 | 0.78 |
| N-NH3 | 0.04 | -0.43 | -0.16 | 0.15 | -0.51 | -0.03 | -0.03 | -0.06 | -0.05 | 0.00 | 0.48 | -0.03 | -0.20 | 0.52 | 1.00 | -0.48 | -0.03 | 0.05 | 0.00 | -0.39 | 0.47 | -0.10 | -0.10 | 0.10 | 0.92 | 0.79 |
| N-NH38 | 0.38 | 0.02 | -0.01 | 0.15 | 0.80 | 0.19 | 0.19 | 0.42 | 0.30 | 0.46 | -0.78 | 0.39 | 0.07 | 0.31 | -0.48 | 1.00 | 0.39 | 0.60 | -0.14 | 0.15 | -0.54 | -0.30 | -0.20 | 0.20 | -0.43 | -0.04 |
| CN | 0.24 | 0.03 | -0.30 | -0.45 | 0.28 | 0.95 | 0.95 | 0.99 | -0.29 | 0.93 | -0.04 | 1.00 | 0.54 | 0.66 | -0.03 | 0.39 | 1.00 | 0.44 | -0.46 | -0.69 | -0.52 | 0.40 | 0.40 | -0.40 | -0.16 | 0.36 |
| SO4 | -0.05 | 0.04 | 0.45 | 0.44 | 0.21 | 0.24 | 0.24 | 0.47 | 0.69 | 0.29 | -0.14 | 0.44 | 0.46 | 0.50 | 0.05 | 0.60 | 0.44 | 1.00 | 0.25 | 0.08 | 0.11 | 0.20 | 0.20 | -0.20 | 0.02 | 0.25 |
| DO | -0.05 | 0.61 | 0.48 | 0.22 | -0.01 | -0.43 | -0.43 | -0.36 | 0.52 | -0.48 | -0.05 | -0.46 | -0.06 | -0.54 | 0.00 | -0.14 | -0.46 | 0.25 | 1.00 | 0.48 | 0.36 | -0.10 | -0.10 | 0.10 | 0.09 | -0.16 |
| DBO | -0.30 | 0.22 | 0.62 | 0.66 | 0.04 | -0.73 | -0.73 | -0.64 | 0.67 | -0.74 | -0.19 | -0.69 | -0.21 | -0.57 | -0.39 | 0.15 | -0.69 | 0.08 | 0.48 | 1.00 | 0.40 | -0.30 | -0.30 | 0.30 | -0.28 | -0.62 |
| Ecl | -0.74 | -0.23 | 0.65 | 0.73 | -0.83 | -0.54 | -0.54 | -0.55 | 0.59 | -0.74 | 0.71 | -0.52 | 0.07 | -0.11 | 0.47 | -0.54 | -0.52 | 0.11 | 0.36 | 0.40 | 1.00 | 0.30 | 0.30 | -0.30 | 0.42 | -0.04 |
| RG | -0.72 | 0.12 | 0.22 | -0.01 | -0.56 | 0.38 | 0.38 | 0.36 | 0.04 | 0.08 | 0.69 | 0.40 | 0.59 | 0.04 | -0.06 | -0.25 | 0.40 | 0.17 | -0.13 | -0.27 | 0.27 | 1.00 | 1.00 | -1.00 | -0.32 | -0.27 |
| RGI | 0.71 | 0.11 | 0.20 | -0.01 | -0.55 | 0.40 | 0.40 | 0.38 | 0.04 | 0.10 | 0.68 | 0.42 | 0.59 | 0.07 | -0.06 | -0.24 | 0.42 | 0.19 | -0.15 | -0.29 | 0.26 | 1.00 | 1.00 | -1.00 | -0.32 | -0.26 |
| Ii | 0.71 | -0.12 | -0.20 | 0.02 | 0.55 | -0.40 | -0.40 | -0.40 | -0.38 | -0.04 | -0.10 | -0.68 | -0.43 | -0.59 | -0.07 | 0.06 | 0.24 | -0.43 | -0.19 | 0.15 | 0.29 | -0.26 | -1.00 | -1.00 | 1.00 | 0.32 | 0.25 |
| MI | 0.17 | -0.48 | -0.03 | 0.14 | -0.36 | -0.11 | -0.11 | -0.17 | 0.01 | -0.07 | 0.31 | -0.16 | -0.15 | 0.45 | 0.92 | -0.43 | -0.16 | 0.02 | 0.09 | -0.28 | 0.42 | -0.30 | -0.30 | 0.30 | 1.00 | 0.83 |
| CAI | 0.47 | -0.45 | -0.31 | -0.13 | 0.01 | 0.34 | 0.34 | 0.34 | -0.18 | 0.48 | 0.04 | 0.36 | -0.01 | 0.78 | 0.79 | -0.04 | 0.36 | 0.25 | -0.16 | -0.62 | -0.04 | -0.30 | -0.30 | 0.30 | 0.83 | 1.00 |

| | | | | | | | | | | | | | | | | | | | | | |
|---|---|---|---|---|---|---|---|---|---|---|---|---|---|---|---|---|---|---|---|---|
| 1.00 | 0.90 | 0.80 | 0.70 | 0.60 | 0.50 | 0.40 | 0.30 | 0.20 | 0.10 | 0.00 | -0.10 | -0.20 | -0.3 | -0.40 | -0.50 | -0.60 | -0.70 | -0.80 | -0.90 | -1.00 |

**Scale**

**Figure 4.** Heat map describing the Pearson's correlation among the analyzed water and sediment parameters of the João Leite stream, Goiás, Brazil. For the parameters that obtained significant

correlations ($p < 0.05$), the numbers are highlighted in red, and those that did not obtain significant correlations are shown in black. TA: room temperature; TU: turbidity; Tc: true color; Alk: total alkalinity; AlK HCO3: Alkalinity HCO3; TH: total hardness; CLO; chlorides; CG: carbon dioxide; TDS: total dissolved solids; N-NO3: nitrate; N-NO2: nitrite; N-NH38: total ammoniacal nitrogen $7.5 < pH \le 8.0$; CN: conductivity; SO4: sulfate; DO: dissolved oxygen; DBO: biochemical oxygen demand; EcI: *Escherichia coli* index; RG: average root growth; RGI: relative growth index; Ii: inhibition index; MI: mitotic index; CAI: index of chromosomal aberrations.

For the analysis of PCA performed, the values of the factorial loads are presented in Table 5. With the PCA of the water multi-indicators, it was possible to explain the 100% variation based on seven factors or components. Factor 1 explained a total variance of the data set of 35.61%, with an Eigen value of 9.26. Factor 1 had a strong and negative influence on the indicators alkalinity total, CLO and CN.

**Table 5.** Principal component analysis of the multiple indicators of the Leite brook, Goiás, Brazil.

| Parameter | Component | | | | | | |
|---|---|---|---|---|---|---|---|
| | Factor 1 | Factor 2 | Factor 3 | Factor 4 | Factor 5 | Factor 6 | Factor 7 |
| TA | −0.45 | −0.85 | −0.16 | 0.05 | −0.24 | 0.01 | −0.04 |
| TW | −0.01 | −0.10 | 0.60 | −0.30 | −0.68 | −0.22 | −0.16 |
| Tu | 0.51 | 0.47 | 0.33 | 0.49 | −0.13 | 0.39 | −0.01 |
| Tc | 0.64 | 0.26 | 0.01 | 0.64 | 0.28 | −0.18 | −0.10 |
| pH | −0.39 | −0.78 | 0.44 | 0.22 | −0.03 | 0.02 | 0.03 |
| Alk | −0.95 | 0.21 | 0.06 | −0.03 | −0.12 | 0.15 | −0.12 |
| Alk HCO3 | −0.95 | 0.21 | 0.06 | −0.03 | −0.12 | 0.15 | −0.12 |
| TH | −0.95 | 0.16 | 0.16 | 0.17 | −0.11 | −0.04 | −0.09 |
| OM | 0.52 | 0.26 | 0.30 | 0.75 | −0.06 | −0.05 | 0.04 |
| CLO | −0.98 | −0.14 | −0.01 | 0.10 | −0.01 | −0.06 | 0.00 |
| CG | 0.16 | 0.86 | −0.38 | −0.25 | −0.07 | 0.04 | −0.14 |
| TDS | −0.96 | 0.21 | 0.11 | 0.15 | −0.01 | −0.05 | −0.05 |
| N-NO3 | −0.38 | 0.64 | 0.30 | 0.26 | −0.10 | 0.53 | 0.09 |
| N-NO2 | −0.62 | 0.12 | −0.48 | 0.55 | 0.23 | −0.08 | −0.06 |
| N-NH3 | 0.02 | 0.18 | −0.91 | 0.19 | −0.22 | −0.20 | −0.05 |
| N-NH38 | −0.35 | −0.41 | 0.52 | 0.58 | 0.28 | −0.15 | 0.05 |
| CN | −0.96 | 0.21 | 0.10 | 0.15 | −0.01 | −0.05 | −0.05 |
| SO4 | −0.23 | 0.24 | 0.25 | 0.87 | −0.19 | −0.18 | 0.04 |
| DO | 0.52 | −0.04 | 0.24 | 0.17 | −0.78 | −0.08 | 0.15 |
| DBO | 0.79 | −0.14 | 0.46 | 0.24 | 0.05 | 0.03 | −0.29 |
| EcI | 0.69 | 0.60 | −0.30 | 0.21 | −0.08 | −0.03 | −0.11 |
| RG | −0.22 | 0.90 | 0.23 | −0.21 | 0.03 | −0.17 | 0.07 |
| RGI | −0.24 | 0.90 | 0.23 | −0.20 | 0.04 | −0.18 | 0.09 |
| Ii | 0.25 | −0.90 | −0.22 | 0.20 | −0.03 | 0.18 | −0.09 |
| MI | 0.11 | 0.00 | −0.91 | 0.28 | −0.27 | 0.11 | 0.01 |
| CAI | −0.42 | −0.12 | −0.78 | 0.39 | −0.21 | 0.01 | 0.09 |
| Eigenvalue | 9.26 | 6.18 | 4.40 | 3.48 | 1.63 | 0.79 | 0.26 |
| Total variance % | 35.61 | 23.76 | 16.93 | 13.38 | 6.26 | 3.04 | 1.01 |
| Cumulative % | 35.61 | 59.38 | 76.31 | 89.69 | 95.95 | 98.99 | 100.00 |

Note: TA: room temperature; TU: turbidity; Tc: true color; Alk: total alkalinity; AlK HCO$_3$: alkalinity HCO$_3$; TH: total hardness; CLO; chlorides; CG: carbon dioxide; TDS: total dissolved solids; N-NO$_3$: nitrate; N-NO$_2$: nitrite; N-NH38: total ammoniacal nitrogen $7.5 < pH \le 8.0$; CN: conductivity; SO4: sulfate; DO: dissolved oxygen; DBO: biochemical oxygen demand; EcI: *Escherichia coli* index; RG: average root growth; RGI: relative growth index; Ii: inhibition index; MI: mitotic index; CAI: index of chromosomal aberrations.

PCA is a multivariate statistical technique that can be used to identify components or factors that explain variations in a system, and is applied to various environmental issues, environmental contamination, and dynamic variation forecasting and monitoring [57]. Study [58] considered that an Eigen value > 1 indicated anthropogenic activities in relation

to metal concentrations in the Subarnarekha River, India. Of the Eigen values found (Table 5), five out of seven were >1, indicating anthropogenic interference in relation to water pollutants. Additionally, the contribution to PCA can be explained by the influence of anthropogenic activity and lithogenic sources in the stream, in addition to the fact that the sites analyzed (sites 2, 3 and 4) are close to highways and roads, which receive a large load of automobiles and the surrounding agricultural area [59,60]. Thus, the PCA outcomes were a result of a mixed source of inputs and pollutants, anthropogenic, industrial and agricultural [61].

Multivariate analysis techniques are very useful in fully characterizing river areas and in helping to indicate risks to the health of the local population [62]. PCA and cluster analysis indicate how to process and reduce the dimensionality of the data, highlight the parameters that have the greatest influence on the qualitative state of the water, and identify clusters; this was observed in this study, indicating that the main influences in this study are true color, BOD and *E. coli* index [63].

## 4. Conclusions

Water quality is an issue raised around the world and is essential for sustaining life. In this study, it was demonstrated—by means of multi-indicators of water quality—that according to the CONAMA classification, current Brazilian legislation and the purpose of the João Leite stream, the water quality meets some parameters (i.e., turbidity, dissolved oxygen, and bacteriological); however, in some samples, the quality is poor or very poor. Samples collected in the rainy season indicated cytotoxicity, probably due to pollutants dragged by the rain into the stream. Mutagenicity effects were not found, but DNA damage was found, suggesting that there are harmful substances in the water samples. The multivariate and cluster analysis indicated that there is anthropogenic influence on this river and that this pollution can occur in different ways (industrial, agricultural or urban) for each sampling site, which confirms that this water system is used for different purposes. Furthermore, significant and strong associations between the various parameters analyzed were demonstrated. As it is a river for leisure and supply, which is used for primary contact recreation, the data presented here are alarming, as the population is directly exposed and various bacterial and diarrheal diseases can occur, causing serious damage to health. Based on the study, the following actions and recommendations are suggested: (I) inspection should take place throughout the riverside area; (II) public policies and public awareness should be focused on; (III) recovery, preservation and maintenance actions should take place; (IV) additional research should be carried out to check for other pollutants, such as checking if there are emerging pollutants in this area or carrying out a risk assessment; and (IV) water must be treated before use, and primary contact recreation must be avoided.

**Author Contributions:** R.P.G.: conceptualization, methodology, data curation, writing—review, writing—original draft preparation, formal analysis and editing. T.R.O.: methodology. A.R.G.: methodology. J.D.G.V.: data curation and supervision. L.C.C.: conceptualization, supervision, data curation and writing—review. All authors have read and agreed to the published version of the manuscript.

**Funding:** This research did not have financial support.

**Data Availability Statement:** Not applicable.

**Acknowledgments:** I offer my sincere thanks to the Companhia de Saneamento de Goiás (SANEAGO), for providing some results. My thanks also go to the Coordination for the Improvement of Higher Education Personnel (CAPES), the State of Goiás Research Support Foundation (FAPEG) and the National Council for Scientific and Technological Development (CNPq).

**Conflicts of Interest:** The author declares that they have no competing interests.

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
