# Peer review of "Multi-Indicators and Evidence of Cytotoxicity—A Case Study of a Stream in Central Brazil"

_water, doi:10.3390/w14192974_

Round 1

Reviewer 1 Report (Previous Reviewer 2)

In this case, the author try to find the relationship between the water parameters and pollutions. I think it is interesting and useful for water quality detection and protection. But I still have some comments as follows.

1) there are only for positions, is it enough or not? I think the author should confirm this and show some evidences.

2) there are too many water parameters data. Are they all necessary? I think some of them are not independence.

3) The final conclusion is too simple.

4) water must be treated before use, and  primary contact recreation must be avoided. I can not agree this item.

So, I think this paper could be published in a local journal.

Author Response

Reviewer #1:

In this case, the author try to find the relationship between the water parameters and pollutions. I think it is interesting and useful for water quality detection and protection. But I still have some comments as follows.

1) there are only for positions, is it enough or not? I think the author should confirm this and show some evidences.

Answer:Yes, only 4 collection points are enough to represent the study area. This can be proven since two other works, already published in the same place, also used only 4 collection points. Their doi are https://doi.org/10.1016/j.scitotenv.2020.141632 and https://doi.org/10.1016/j.enmm.2022.100688

2) There are too many water parameters data. Are they all necessary? I think some of them are not independence.

Answer: These unregulated parameters were studied because in the literature it indicates that if they have alterations they can cause damage, mainly related to cytotoxic and mutagenic data. No further changes were detected.

3) The final conclusion is too simple.

Answer: divergence among reviewers, the other reviewer praised the conclusion. Therefore, it has not been modified.

4) water must be treated before use, and  primary contact recreation must be avoided. I can not agree this item.

Answer: This statement was given since some parameters tested are not in accordance with Brazilian legislation for water use (consumption) and primary recreation contact. Since the parameters do not agree, for these uses it should be avoided.

references have been updated

Reviewer 2 Report (New Reviewer)

-          The abstract section should be corrected, and the results obtained from the study should be expressed in it.

-          The introduction is very short.

-           In this section, it is better to refer to similar studies that have been done with this topic. The types of monitoring methods should also be stated.

-          Explanations about the studied area should be given in the introduction.

-          Evaluation of the cytotoxic effect of mixed pollutants, genetic damage should be explained more in the introduction.

-          In the materials and methods section, the title of the second part is wrongly written, sampling location. That should be corrected. In this section, the number of collected samples and their volume should be stated.

-          Table 1 is very confusing. It is better to draw a table for physical parameters, a table for chemical and a separate table for biological.

-          Many not regulated items are mentioned in the table. Please explain the reason.

-          In the discussion section, more articles should be compared with the present study.

-          The conclusion is well stated. It is better to use some of the sentences of this part in the abstract of the article.

Author Response

Reviewer #2:

The abstract section should be corrected, and the results obtained from the study should be expressed in it.

Answer: summary modified according to the reviewer's last suggestion.

-          The introduction is very short.

Ansswer: modified introduction

-           In this section, it is better to refer to similar studies that have been done with this topic. The types of monitoring methods should also be stated.

Answer: added in introduction.

-          Explanations about the studied area should be given in the introduction.

Answer: The sentence was taken from the material and methods and added to the introduction.

-          Evaluation of the cytotoxic effect of mixed pollutants, genetic damage should be explained more in the introduction.

Answer: added in introduction.

-          In the materials and methods section, the title of the second part is wrongly written, sampling location. That should be corrected. In this section, the number of collected samples and their volume should be stated.

Answer: adjusted.

-          Table 1 is very confusing. It is better to draw a table for physical parameters, a table for chemical and a separate table for biological.

Answer: separate tables.

-          Many not regulated items are mentioned in the table. Please explain the reason.

Answer: although the parameters are not regulated, the literature indicates that if they have alterations they can cause damage, mainly related to cytotoxic and mutagenic data. No further changes were detected.

-          In the discussion section, more articles should be compared with the present study.

Answer: Complemented.

-          The conclusion is well stated. It is better to use some of the sentences of this part in the abstract of the article.

Answer: suggestion answered.

references have been updated

Round 2

Reviewer 1 Report (Previous Reviewer 2)

In this case, I think it is fine after the explaination and correction. But I can't agree the answer of item 3. No further comments.

Author Response

This manuscript is a resubmission of an earlier submission. The following is a list of the peer review reports and author responses from that submission.

Round 1

Reviewer 1 Report

Manuscript Number: water-1149502

Title: Multi-Indicators, Evidence of Cytotoxicity / Genotoxicity , Health Risk Assessment in an Aquatic System

Authors: Raylane P. Gomes, Thais R. Oliveira, Aline G. Rodrigues, José D. G. Vieira, Thiago L. Rocha, Lilian C. Carneiro

Current manuscript is not acceptable for publication.

As critical issue, analytical methods and data treatment are unclear in this study. In other word, reliability of the study results is unknown.

There are so many explanations before approach and almost all explanations are not necessary. Also, the cited references are no need. 136 citations are too many.

English in the manuscript should be revised by native speakers.

Reviewer 2 Report

Water quality is an issue raised across the globe and is essential for maintaining life. So how to assess the water contamination and evaluate the multi-indicators is becoming a challenging topic. This work provides basic data for developing environmental prevention, control and management policies. In this case, it is valuable for readers. But within a local data, the universality is uncertain. I am also having some other comments as follows.

  1. Most of all the indicators are reported in references, so the innovation is poor.
  2. Only within four samples, the final decision is right or not?
  3. And also, the coping strategies are not well reflected in the paper.

As a comment, the government should pay much attention for public health.